## EDITORIAL

## Physiological Considerations for Maximum Indoor Temperatures

Thomas E J Addison[1] ,
Daniel Mark Blake[2], Paul Coleman[2] ,
Florence Lock[2], Emily Loud[2],
Andrew Mackenzie[1], Shania Pande[1]
and Mike Tipton[1,3]

[1] *The Physiological Society, 30 Farringdon Lane, London, EC1R 3AW, United Kingdom*
[2] *UK Health Security Agency, London, United Kingdom*
[3] *Extreme Environments Laboratory, University of Portsmouth, Cambridge Road, Portsmouth, PO1 2ER, United Kingdom*

Email: taddison@physoc.org

The peer review history is available in the Supporting Information section of this article (https://doi.org/10.1113/JP289174#support-information-section).

### Introduction

**Heat, safety and productivity.** The past three years (2022, 2023 and 2024) have been in the top 4 hottest years on record in the UK (Hawkins 2025)[i]. The 10 warmest years recorded in the UK have all been since 2003 and the UK recorded its first air temperatures of above 40°C in 2022. During the extreme heat of summer 2022, there were an estimated 2,985 all-cause excess deaths, mostly among vulnerable populations which are likely to increase as a proportion of the overall UK population as the population ages. Exploring the evidence-base for potentially establishing maximum temperature thresholds for indoor environments in the UK therefore is of increasing importance with hotter summers becoming more frequent and temperatures more extreme.

As a consequence of anthropogenic climate change and the health hazards it has exacerbated, we are seeing an increase in development of guidance and legislation focusing on maximum temperature thresholds within indoor working environments. The possibility of establishing maximum indoor temperature thresholds in the UK for a variety of vulnerable groups and settings is a complex challenge but one that public health professionals are increasingly grappling with.

**Existing legislation and guidance related to extreme temperatures in the UK.** Since 1992, the Approved Code of Practice associated to the Workplace (Health, Safety and Welfare) Regulations have recommended that an indoor working space should not go below 16°C, or 13°C if employees are engaged in physically active tasks. This recognises that prolonged exposure to low temperatures increases the health risks to workers through not only hypothermia but also musculoskeletal injuries and respiratory conditions.

No comparable legislation on maximum indoor temperatures exists from the Health and Safety Executive (HSE). Employers are required to maintain a "reasonable" indoor temperature based on the work activity and environmental conditions, with the focus on providing a comfortable working environment for employees. General guidance on maximum indoor temperature thresholds are included in the UKHSA's *Adverse Weather and Health Plan*, supporting evidence document (SED) and from trade unions.

### Workshop development

There is a need to include physiological considerations when assessing existing and evolving evidence on the health impacts of heat and ability of individuals to mitigate these impacts in order to inform maximum permissible indoor temperature guidance.

The Physiological Society and UK Health Security Agency (UKHSA) held a joint online workshop, 'Physiological Considerations for Maximum Indoor Temperatures'. It brought together 45 delegates from around the world, covering topics from existing evidence and current research efforts, to improving the interactions between thermophysiologists and executive government agencies with an interest in developing evidence and guidance in this area.

The workshop's breakout rooms were structured around exploring three key areas: evidence, obstacles and research gaps for establishing maximum indoor temperature thresholds; identification of vulnerable groups and their inclusion in trial criteria; and future areas for intervention, prioritisation and collaboration.

### Workshop key findings

The comments from the workshop participants highlighted a number of key issues related to physiological considerations for establishing maximum indoor temperatures.

1. A fixed, 'binary' temperature was not considered to be suitable, as variations in response to temperature play a significant role in normal physiology, humidity and stress responses can differ depending on those variations.
2. The number of variables within current heat stress indices could be considered a barrier to the development and implementation of maximum heat thresholds for different settings, groups and physical activity levels. Simpler categories (e.g., risk categories 1–5) to ensure accessibility to the public and provide the nuance that reflects the additional vulnerabilities that might increase an individual's risk level.
3. Thermoregulatory challenges for older adults are well-reported. However, other potentially vulnerable individuals, such as those with mental health conditions, neurological disorders, peri- and post-menopausal women, those in care homes or young children, lack prioritisation and research dedicated to them.
4. While older adults are often considered to be a homogenous group, there is significant variation in health and activity levels among them. Physical activity and appropriate hydration is key for healthy aging, and interventions should also promote staying active to build resilience, not just address acute high indoor temperatures.
5. Certain settings are less able to benefit from sustainable cooling techniques and others will require specific interventions for effective cooling. This is either because the specific intervention is less

---

[i]Hawkins, E. (2025). 2024: UK's fourth warmest year on record highlights urgency for climate action., National Centre for Atmospheric Science, January 2025, https://ncas.ac.uk/2024-uks-fourth-warmest-year-on-record-highlights-urgency-for-climate-action/

effective among the majority of people (e.g. fans are less effective in some vulnerable groups); there are limits on airflow owing to security, infection control and building design (e.g. prisons or ICUs), or workers can't rest or cool down owing to scheduling or personal protective equipment requirements.

6. There is a lack of clarity on defining clear endpoints for mitigation techniques: participants noted that it is important to define what the endpoint for studies should be since strategies will differ depending on the goal. Options include discomfort, cardiovascular disease (CVD), high body temperatures, adverse health outcomes, or mortality.

## Workshop recommendations

In order to deliver progress on the key findings from the workshop, the participants suggested the following recommendations:

1. Promote greater interdisciplinary collaboration on identifying barriers to implementation of physiological research, actions and interventions on heat across academia, government and health and social care

2. Fill research gaps on building resilience to heat, particularly defining inclusion criteria for key vulnerable groups and those most at risk of extreme heat.

3. Explore a range of risk categories and related actions for individuals, businesses / specific settings and the government.

4. Better communicate risks caused by heat and promote options for low-carbon cooling solutions (which are often simple, effective and cheap to implement).

5. Develop clear endpoints for trials into extreme heat thresholds for consistency across research groups in different institutions.

6. Promote existing examples of good practice from other parts of the world on building heat resilience.

## Workshop conclusions

The evidence for setting a maximum indoor temperature threshold is not in place yet. Several challenges hinder the establishment of a clear threshold. Key issues include defining what specific outcome (e.g., discomfort, health changes, or cognitive impacts) the threshold should

aim to prevent, as well as the complexity of individual responses to heat, which vary based on factors like age, pre-existing health conditions, lack of acclimatisation, and activity level. This is particularly true in settings where different groups of people have differing levels of metabolic heat production or are less able to rely on behavioural interventions e.g. care homes. There is also a concern that setting a threshold might lead to people ceasing work once the temperature reaches that limit, potentially disrupting productivity or existing heat-related protocols in specific workplaces; or a focus on maintaining temperatures just below this threshold, rather than a wider consideration of healthy indoor working environments. Moreover, thresholds would need to account for various factors like humidity and individual conditions (e.g., vulnerable groups), making a one-size-fits-all solution difficult.

Participants suggested that a general range could work for the majority of the population, but more research is needed to refine this based on specific groups and settings. A personalised, graded approach would be more effective, taking into account the variability of responses to heat. Additionally, public health messaging and education are essential to ensure people understand and act on any temperature guidelines. Overall, while there was agreement on the need for appropriately evidenced interventions, these require further development, particularly in how they would be applied and communicated across different environments.

## Additional information

### Competing interests

No competing interests declared.

### Author contributions

Thomas Addison: Conception or design of the work; Drafting the work or revising it critically for important intellectual content; Final approval of the version to be published; Agreement to be accountable for all aspects of the work Daniel Blake: Conception or design of the work; Drafting the work or revising it critically for important intellectual content; Final approval of the version to be published; Agreement to be accountable for all aspects of the work Paul Coleman: Conception or design of the work; Drafting the work or revising it critically for important intellectual content; Final approval of the version to be published; Agreement to be accountable for all aspects of the work Florence Lock: Conception or design of the work; Drafting the work or revising it critically for important intellectual content; Final approval of the version to be published; Agreement to be accountable for all aspects of the work Emily Loud: Conception or design of the work; Drafting the work or revising it critically for important intellectual content; Final approval of the version to be published; Agreement to be accountable for all aspects of the work Andrew Mackenzie: Conception or design of the work; Drafting the work or revising it critically for important intellectual content; Final approval of the version to be published; Agreement to be accountable for all aspects of the work Shania Pande: Conception or design of the work; Drafting the work or revising it critically for important intellectual content; Final approval of the version to be published; Agreement to be accountable for all aspects of the work Mike Tipton: Conception or design of the work; Drafting the work or revising it critically for important intellectual content; Final approval of the version to be published; Agreement to be accountable for all aspects of the work.

### Funding

The authors received no specific funding for this work. Some co-authors of this paper were funded by the National Institute for Health and Care Research (NIHR) Health Protection Research Unit (HPRU) in Environmental Change and Health (NIHR 200909), a partnership between UK Health Security Agency and the London School of Hygiene and Tropical Medicine (LSHTM), in collaboration with University College London and the Met Office. The views expressed are those of the co-author(s) and not necessarily those of the NIHR, UK Health Security Agency, London School of Hygiene and Tropical Medicine, University College London, the Met Office or the Department of Health and Social Care.

### Acknowledgements

This article has been simultaneously co-published in *The Journal of Physiology* and *PLOS Climate*. The articles are identical except for minor stylistic and spelling differences in keeping with each journal's style. Either citation can be used when citing this article.

### Keywords

climate change, occupational health, policy, public health, temperature, thermoregulation

## Supporting information

Additional supporting information can be found online in the Supporting Information section at the end of the HTML view of the article. Supporting information files available:

**Peer Review History**

