## [Peer Review History · The Journal of Physiology]

Physiological Considerations for Establishing Maximum Indoor Temperatures Guidance

Thomas E J Addison, Daniel Mark Blake, Paul Christopher Coleman, Florence Lock, Emily Loud, Andrew Mackenzie, Shania Pande, and Mike Tipton

DOI: 10.1113/JP289174

Corresponding author(s): Thomas Addison (taddison@physoc.org)

Review Timeline:

Submission Date:

08-May-2025

Accepted:

08-May-2025

Senior Editor: Senior Editor

Reviewing Editor: Reviewing Editor

Transaction Report:

Dear Tom,

Re: JP-E-2025-289174 "Physiological Considerations for Establishing Maximum Indoor Temperatures Guidance" by Thomas E J Addison, Daniel Mark Blake, Paul Christopher Coleman, Florence Lock, Emily Loud, Andrew Mackenzie, Shania Pande, and Mike Tipton

We are pleased to tell you that your paper has been accepted for publication in The Journal of Physiology.

TRANSPARENT PEER REVIEW POLICY: To improve the transparency of its peer review process, The Journal of Physiology publishes online the peer review history of all articles accepted for publication as supporting information. Readers will have access to decision letters, including Editors' comments and referee reports, for each version of the manuscript, as well as any author responses to peer review comments. Referees can decide whether or not they wish to be named on the peer review history document.

Authors should note that it is too late at this point to offer corrections prior to proofing. Major corrections at proof stage, such as changes to figures, will be referred to the Editors for approval before they can be incorporated. Only minor changes, such as to style and consistency, should be made at proof stage. Changes that need to be made after proof stage will usually require a formal correction notice.

Best wishes

Diana
The Journal of Physiology

P.S. - You can help your research get the attention it deserves! Check out Wiley's free Promotion Guide for best-practice recommendations for promoting your work at www.wileyauthors.com/eeo/guide. You can learn more about Wiley Editing Services which offers professional video, design, and writing services to create shareable video abstracts, infographics, conference posters, lay summaries, and research news stories for your research at www.wileyauthors.com/eeo/promotion.

IMPORTANT NOTICE ABOUT OPEN ACCESS: To assist authors whose funding agencies mandate public access to published research findings sooner than 12 months after publication, The Journal of Physiology allows authors to pay an Open Access (OA) fee to have their papers made freely available immediately on publication.

You can check if your funder or institution has a Wiley Open Access Account here: <https://authorservices.wiley.com/author-resources/Journal-Authors/licensing-and-open-access/open-access/author-compliance-tool.html>.